# Dynamic Self-training Framework for Graph Convolutional Networks

## Abstract

Graph neural networks (GNN) such as GCN, GAT, MoNet have achieved state-of-the-art results on semi-supervised learning on graphs. However, when the number of labeled nodes is very small, the performances of GNNs downgrade dramatically. Self-training has proved to be effective for resolving this issue, however, the performance of self-trained GCN is still inferior to that of G2G and DGI for many settings. Moreover, additional model complexity make it more difficult to tune the hyper-parameters and do model selection. We argue that the power of self-training is still not fully explored for the node classification task. In this paper, we propose a unified end-to-end self-training framework called *Dynamic Self-traning*, which generalizes and simplifies prior work. A simple instantiation of the framework based on GCN is provided and empirical results show that our framework outperforms all previous methods including GNNs, embedding based method and self-trained GCNs by a noticeable margin. Moreover, compared with standard self-training, hyper-parameter tuning for our framework is easier.

## 1 Introduction

Graphs or networks can be used to model any interactions between entities such as social interactions (Facebook, Twitter), biological networks (protein-protein interaction), and citation networks. There has been an increasing research interest in deep learning on graph structured data, e.g., (Bruna et al., 2014; Defferrard et al., 2016; Monti et al., 2017; Kipf & Welling, 2017; Hamilton et al., 2017; Velickovic et al., 2018; Tang et al., 2015; Perozzi et al., 2014).

Semi-supervised node classification on graphs is a fundamental learning task with many applications. Classic methods rely on some underly diffusion process to propagate label information. Recently, *network embedding* approaches have demonstrate outstanding performance on node classification (Tang et al., 2015; Grover & Leskovec, 2016; Bojchevski & Günnemann, 2018). This approach first learns a lower-dimensional embedding for each node in an unsupervised manner, and then the embeddings are used to train a supervised classifier for node classification, e.g., logistic regression or multi-layer perceptron (MLP). Graph neural networks (GNN) are semi-supervised models and have achieved state-of-the-art performance on many benchmark data sets (Monti et al., 2017; Kipf & Welling, 2017; Velickovic et al., 2018). GNNs generalize convolution to graph structured data and typically have a clear advantage when the number of training examples is reasonably large. However, when there are very few labeled nodes, GNNs is outperformed by embedding based method (as shown by our experimental results), e.g., G2G from (Bojchevski & Günnemann, 2018) and DGI from (Veličković et al., 2019).

To overcome this limitation of GCNs (Kipf & Welling, 2017), Li et al. (Li et al., 2018) propose to apply self-training and co-training techniques (Scudder, 1965). The idea of these techniques is to augment the original training set by adding in some unlabeled examples together with their label predictions. Such "pseudo-label" information is either from the base model trained on the original training set (self-training) or another learning algorithm (co-training). The results from (Li et al., 2018) demonstrate the effectiveness of co-training and self-training. However, among the four variants implemented in (Li et al., 2018), there is not a single one that achieves the best performance across different settings; and from our experiments, G2G and DGI outperforms all the four variants when the number of labels from each class is less than 10. There are clear restrictions in prior self-training approaches. First, the pseudo-label set is incremental only, i.e., after an unlabeled

example is added to the training set, it will never be deleted and its pseudo-label will never change even if its prediction and/or the corresponding margin has changed drastically. Secondly, all the pseudo-labels are considered equal, although they may have very different classification margins. Furthermore, it introduces extra hyper-parameters such as the number of unlabeled nodes to be added into the training set and the total number of self-training iterations. The performance gain is sensitive to such parameters and their optimal values may differ for different data sets and label rates (Buchnik & Cohen, 2018).

To fully understand and explore the power of self-training on the node classification task, we propose a novel self-training framework, named *Dynamic Self-training*, which is general, flexible, and easy to use. We provide a simple instantiation of the framework based on GCN (Kipf & Welling, 2017) and empirically show that it outperforms state-of-art methods including GNNs, self-trained GCN (Li et al., 2018), and embedding based methods. Our framework has the following distinguishing features compared with (Li et al., 2018; Buchnik & Cohen, 2018).

1. We augment the training set and recalculate the pseudo-labels *after each epoch*. So the number self-training iterations is the same as the number of epochs and the pseudo-label assigned to an unlabeled example may change during the training process.

2. In stead of inserting a fixed number of new pseudo-labels with highest margin in each iteration, we use a threshold-based rule, i.e., insert an unlabeled node if and only if its classification margin is above the threshold.

3. The pseudo-label set is dynamic. When the margin of an unlabeled node is above the threshold, we activate it by adding it to the loss function, but if the margin of this node becomes lower than the threshold in a later epoch, we will deactivate it.

4. We assign a (dynamic) personalized weight to each active pseudo-label proportional to its current classification margin. The total pseudo-label loss is thus the weighted sum of losses corresponds to all pseudo-labels.

## 2 PRELIMINARIES

### 2.1 GRAPH NOTATION AND PROBLEM DEFINITION

In the problem, we are given an undirected graph with node attributes $G = (V, E, X)$, where $V$ is the vertex set, $E$ is the edge set. Here, $X$ is the feature matrix, the $i$-th row of which, denoted as $x_i$, is the feature vector of node $i$. We assume each node belongs to exactly one class and use $y_i$ to denote the class label of the $i$-th node. The aim is to design learning algorithms to predict the labels of all nodes based on the labels of a small set of training nodes provided in the beginning. We use $\mathcal{N}_k(i)$ to denote the set of nodes whose distance to node $i$ is at most $k$. $\mathcal{L} \subset V$ is the set of labeled nodes and $\mathcal{U} = V \setminus \mathcal{L}$ is the set of unlabeled nodes.

### 2.2 GRAPH CONVOLUTIONAL NETWORKS

GCN introduced in (Kipf & Welling, 2017) is a graph neural network model for semi-supervised classification. GCN learns the representations of each node by iteratively aggregating the embeddings of its neighbors. Specifically, GCN consists of $L > 0$ layers each with the same propagation rule defined as follows. In the $l$-th layer, the hidden representations $H^{(l-1)}$ are averaged among one-hop neighbors as:

$$H^{(l)} = \sigma(\tilde{D}^{-\frac{1}{2}} \tilde{A} \tilde{D}^{-\frac{1}{2}} H^{(l-1)} W^{(l)}). \tag{1}$$

Here, $\tilde{A} = A + I_n$ is the adjacency matrix of $G$ after adding self-loops ($I_n$ is the identity matrix), $\tilde{D}$ is a diagonal matrix with $\tilde{D}_{ii} = \sum_j \tilde{A}_{ij}$, $W^{(l)}$ is a trainable weight matrix of the $l$-th layer, and $\sigma$ is a nonlinear activation function; $H^{(l)} \in \mathbb{R}^{n \times d_l}$ denotes hidden feature matrix of the $l$-th layer and $H^{(0)} = X$ and $f_i = H_i^{(L)}$ represents the output of $i$-th node.

We use $l(y_i, f_i)$ to denote the classification loss of node $i$, which is typically the cross entropy function. Thus, loss function used by GCN is of the form:

$$L = \sum_{i \in \mathcal{L}} l(y_i, f_i) \tag{2}$$

For a $k$-layer GCN, the receptive field of each training example is its order-$k$ neighborhood. When there are only few training samples, we need to increase the number of layers in order to cover most of the unlabeled nodes. However, deeper GCN will cause the problem of *over-smoothing*, i.e., critical features of the vertices may be smoothed through the iterative averaging process, which makes nodes from different class indistinguishable (Xu et al., 2018; Li et al., 2018).

### 2.3 SELF TRAINING

Recently (Li et al., 2018) apply self-training to overcome these limitations of GCNs. Self-training is a natural and general approach to semi-supervised learning, which is particularly well-motivated in the context of node classification (Buchnik & Cohen, 2018; Li et al., 2018). Assume we have a base model/algorithm for the learning problem, which takes as input a set of labeled examples and makes predictions for other examples. Typically, for each unlabeled node, the base algorithm will also return an associated margin or confidence score. The self-training framework trains and applies the base model in rounds, where at the end of each round, the highest-confidence predictions are converted to become new labeled examples in the next round of training and prediction. Thus, the receptive fields of all the labeled nodes increases and will eventually cover the entire graph, which resolve the issue of GCNs without adding more layers.

## 3 OUR METHOD

### 3.1 A GENERALIZED SELF-TRAINING FRAMEWORK

---
**Algorithm 1:** Dynamic Self-training Framework

---
1 Generate initial parameter $\theta^0$ for model $f(\cdot, \cdot)$, and the initial confidence score vector $S_V$ .
2 **for** each epoch $t = 1, 2, ..., T$ **do**
3     **Compute prediction** $f_V \leftarrow f(G, \theta^{t-1})$
4     **Update confidence score** $S_V \leftarrow \mathcal{UC}(f_V)$.
5     **Update model parameter** by confidence score. $\theta^t \leftarrow \mathcal{UP}(f_V, S_V, f)$
6     **if** stopping criteria is met **then**
7         Break
8     **end**
9 **end**

---

Sun et al. (Sun et al., 2019) proposed Multi-stage Training Framework as generalization for self-training method in (Li et al., 2018). Inspired by this, we propose a more generalized end-to-end self-training framework named *Dynamic Self-training Framework* shown in algorithm 1. Instead of operating on data split, we maintain a confidence score in each iteration. There is no specified training stages here, but we update the confidence value for each unlabeled node after every epoch.

Consider the original model $f(\cdot, \cdot)$ as a forward predicting function with backward trainable parameters. The graph data $G$ and the trainable parameters $\theta^t$ is the input of this function, and the output of this model is collected into $f_V \in \mathbb{R}^{n \times C}$, where $f_v$ denotes the output vector (before assigned with label) of node $v \in V$, and $C = d_L$ is the number of classes. Then we construct the confidence score vector $S_V \in \mathbb{R}^n$ from the model output $f_v$ using a function $\mathcal{UC}$, which can be instantiated in many forms. For example, Algorithm 2 illustrates how standard multi-stage self-training GCN implement this part. Finally we update the model parameters using a specified algorithm such as gradient descent, where the confidence score vector plays a role. The confidence score participates in the parameter updating process in an end-to-end manner. An example of this part can be seen in section 3.3.

### 3.2 PSEUDO LABEL METHOD

Define the pseudo label $\tilde{y}_i \in \mathbb{R}^{d_L}$ of $i$-th node which satisfies :

$$\tilde{y}_{ij} = \begin{cases} 1 & \text{if } j = \arg\max_{j'} f_{ij'} \\ 0 & \text{otherwise} \end{cases} \tag{3}$$

---

**Algorithm 2: Update confidence score** for Multi-stage Self-training GCN

---

1    **if** the stage is currently switched **then**
2      **for** each class $k$ **do**
3        Find the top $m$ vertices $v$ in $f_V$ and $v \in \mathcal{U}$
4        Change the value of $v$ in $S_V$ to 1
5      **end**
6      return $S_V$
7    **end**

---

(Lee, 2013) introduced a pseudo label version of semi-supervised losses:

$$L = \sum_{i \in \mathcal{L}} l(y_i, f_i) + \lambda \sum_{i \in \mathcal{U}} l(\tilde{y}_i, f_i), \tag{4}$$

where $\lambda = \frac{n}{n'}\gamma$, $n = |\mathcal{L}|$, $n' = |\mathcal{U}|$, $\gamma \in \mathbb{R}$ is a hyper-parameter and the additive term $\sum_{i \in \mathcal{U}} l(\tilde{y}_i, f_i)$ is the pseudo label loss. Here, $\lambda$ measures how much the pseudo label term influence the training process. This is equivalent to Entropy Regularization for classification problems (Lee, 2013).

### 3.3   SOFT LABEL CONFIDENCE

In standard multi-stage self-training methods, a node just has two states: in the training set or not, which corresponds to binary-valued confidences $\{0, 1\}$; and in most cases, if a node is added in training set, it will be kept there. This simple setting hinders learning in some cases. For instance, if the classifier puts a wrongly labeled node into the training set, which is of high possibility in preliminary training epochs, it will persistently learn wrong knowledge from this node. Worse still, another wrongly adding is more possible. This negative feedback loop may contribute to a extremely poor classifier. Moreover, original labeled nodes and added nodes in the training are treated equally, which is too restricted and may harm the learning; explicitly distinguishing them in the training process could be beneficial. To resolve these problems, we introduce a mechanism named *Soft Label Confidence* as the confidence updating component in algorithm 1, which computes a personalized confidence value for each node, and the training set is dynamically changing except the ground truth labels. Based on the pseudo label loss (4), we propose the loss wrapped by soft label confidence:

$$L = \sum_{i \in \mathcal{L}} l(y_i, f_i) + \lambda \sum_{i \in \mathcal{U}} \alpha(f_i) l(\tilde{y}_i, f_i). \tag{5}$$

Here $\alpha$ is a function mapping from $\mathbb{R}^{d_L}$ to $\mathbb{R}$, defined as *confidence function*. While there are other possible choices for $\alpha$, in our method we adopt a threshold based function:

$$\alpha(f_i) = \frac{1}{n'_{c^i}} \max(\text{ReLU}(f_i - \beta \cdot \mathbf{1})), \tag{6}$$

Here $\beta \in (0, 1)$ is a hyper-parameter as threshold, $n'_{c^i}$ denotes the number of nodes whose pseudo label belongs to class $c^i$, $c^i$ is the class which $i$-th node's pseudo label belongs to, and $\mathbf{1}$ is the all 1 vector. We introduce $n'_{c^i}$ here to balance the categories of pseudo labels, because pseudo labels could be initially extremely unbalanced and lead to a poor classifier in practice.

Although $\alpha(f_i)$ depends on $f_i$, and thus a function of network's weights, we will block the flow of gradient through $\alpha(f_i)$ for the following reasons: Firstly, confidence function is non-differentiable in most cases. Secondly, if we allow the gradient to flow through $\alpha(f_i)$, the optimizer may tend to find a solution that satisfies $\max(f_i) < \beta, \forall i \in V$, since for such a solution, $\alpha(f_i) = 0$ for all $i$ and the pseudo label loss is zero, which does no good to self-supervised learning. So we use the following way to compute the gradient:

$$\frac{\partial L}{\partial W^l_{s,t}} = \sum_{i \in \mathcal{L}} \frac{\partial l(y_i, f_i)}{\partial W^l_{s,t}} + \lambda \sum_{i \in \mathcal{U}} \alpha(f_i) \frac{\partial l(\tilde{y}_i, f_i)}{\partial W^l_{s,t}} \tag{7}$$

### 4   RELATED WORK

**Graph Convolutional Network**   The work of GNNs seeks generalizations of the convolution operator to graph structured data. One way to do this is to apply convolution in the spectral domain,

where the eigenvectors of the graph Laplacian are considered as the Fourier basis (Bruna et al., 2014; Henaff et al., 2015; Defferrard et al., 2016; Kipf & Welling, 2017). Such spectral methods learns hidden layer representations that encode both graph structure and node features simultaneously. Kipf and Welling (Kipf & Welling, 2017) simplify previous spectral techniques by restricting the propagation to a 1-hop neighborhood in each layer. (Chen et al., 2018) propose fast GCNs, which improves the training speed of the original GCN. GAT of (Velickovic et al., 2018) allows for assigning different importances to nodes of the same neighborhood via attention mechanisms. (Xu et al., 2018) introduce JK networks, which adjust the influence radii of each node adaptively. Another direction that generalizes convolutions to graph structured data, namely non-spectral approaches, define convolutions directly in the spatial domain (Duvenaud et al., 2015; Atwood & Towsley, 2016; Monti et al., 2017). Such methods are easier to be adapted to do inductive learning (Hamilton et al., 2017; Velickovic et al., 2018; Bojchevski & Günnemann, 2018). However, few-shot learning remains a challenge for this class of methods.

**Label Propagation**    Unlike GNNs, which propagate node representations, the classic Label Propagation (LP) method (Zhu et al., 2003) iteratively propagates (soft) labels. More specifically, in each iteration, each unlabeled node obtains a new soft label that is the aggregation of the soft labels from the previous iteration of its neighbors. The key to LP is to design an effective propagation rule; for some propagation rules, the algorithm may not converge and/or the accuracy may not improve over iterations. Thus, one often needs to specify a stopping criteria and a validation set for model selection. LP can also be used as the base algorithm in the self-training framework.

**Self-training**    Self-training is a natural and general approach to semi-supervised learning (Scudder, 1965) and has been widely used in the NLP literature. Self-training is used by (Yarowsky, 1995; Hearst, 1991) for word sense disambiguation. (Riloff et al., 1999) used self-training in the form of bootstrapping for information extraction and later for learning subjective nouns. (Riloff et al., 2003) with (Nigam et al., 2000) using EM for text classification. Self-training has been used for object recognition (Rosenberg et al., 2005; Zhou et al., 2012). (McClosky et al., 2006; 2008; Huang & Harper, 2009; Sagae, 2010) shows how effective can self-training be in parsing. (Wang et al., 2007; Huang et al., 2009; Qi et al., 2009) introduce self-training techniques to part of speech tagging, and (Kozareva et al., 2005; Liu et al., 2013a) adopt self-training in named entity recognition. (Van Asch & Daelemans, 2016; Drury et al., 2011; Liu et al., 2013b) used self-training in sentiment classification. Recently, self-training has also been successfully applied on node classification. Li et al. (Li et al., 2018) study self-training GCNs; Buchnik and Cohen (Buchnik & Cohen, 2018) mainly consider the effect self-training for diffusion-based techniques. In pseudo-label method of (Lee, 2013), for unlabeled data, their pseudo-labels are recalculated every weights update. However, they don't assign weight to each unlabeled data.

As for the self-training algorithm itself, (Chen et al., 2011) shows that selecting highly confident instances with a pre-defined threshold may not perform well. (McClosky et al., 2006) produce a ranked list of n-best predicted parses and selected the best one. (Rosenberg et al., 2005) shows that a training data selection metric that is defined independently of the detector greatly outperforms a selection metric based on the detection confidence generated by the detector. (Zhou et al., 2012) suggests that selecting more informative unlabelled data using a guided search algorithm can significantly improve performance over standard self-training framework. Most recently, (Levatić et al., 2017) proposed proposed an algorithm to automatically select appropriate threshold.

**Network Embedding**    Node classification is also one of the main applications of network embedding methods, which learns a lower-dimensional representation for each node in an unsupervised manner, followed by a supervised classifier layer for node classification (Perozzi et al., 2014; Tang et al., 2015; Grover & Leskovec, 2016; Wang et al., 2016; Bojchevski & Günnemann, 2018). A recent work of (Bojchevski & Günnemann, 2018) proposes Graph2Gauss. This method embeds each node as a Gaussian distribution according to a novel ranking similarity based on the shortest path distances between nodes. A distribution embedding naturally captures the uncertainty about the representation. DGI (Veličković et al., 2019) is an embedding method based on GCNs, the unsupervised objective of which is to maximize mutual information. The work of Embedding approaches achieve competitive performance in node classification tasks, while the learned representations also prove to be extremely useful for other downstream applications.

## 5 EVALUATION

### 5.1 DATASET

We conduct the evaluation on four benchmark citation datasets: Cora, Citeseer, Pubmed (Sen et al., 2008), and Core-full (Bojchevski & Günnemann, 2018). Each of these four datasets is undirected graph with node feature. Each node is a document and the edges denote the citation relationship; the feature of a node is the bag-of-words representation of the document. The number of layers in GCN is two by default, and thus the receptive field of each labeled node is its order-2 neighborhood. We measure the fraction of nodes which is covered by the 2-hop neighbors of all labeled nodes, i.e., $|\bigcup_{s \in \mathcal{S}} \mathcal{N}_2(s)|/|V|$, where $\mathcal{S}$ is the set of labeled nodes randomly sampled from $V$. Here we report the 2-hop coverage ratio on the four datasets when the label rates are 1% and 0.5% respectively. We summarize the information of datasets in Table 1.

Table 1: Summary of datasets

|  | Cora | Citeseer | Pubmed | Cora-full |
|---|---|---|---|---|
| # of Nodes | 2708 | 3327 | 19717 | 18703 |
| # of Edges | 5429 | 4732 | 44338 | 81124 |
| # of Features | 1433 | 3703 | 500 | 8710 |
| # of Classes | 7 | 6 | 3 | 67 |
| Coverage(0.5%) | 14.78% | 6.64% | 21.58% | 27.19% |
| Coverage(1%) | 24.78% | 12.14% | 34.60% | 47.42% |

### 5.2 EXPERIMENT SETTINGS

We evaluate models on semi-supervised node classification tasks with varying label rates. Instead of evaluating on a fixed data split as in (Kipf & Welling, 2017; Velickovic et al., 2018), we mainly consider random splits as (Li et al., 2018) does. In detail, for a given label rate, we randomly generate 100 different splits on each dataset. In each split, there is a labeled set with prespecified size for training, and in this set each class contains the same number of labeled nodes. As in (Li et al., 2018), we don't use a validation set, and all the remaining nodes will be used for testing. For simplicity, we will refer to a task in the form of dataset-$l$, where $l$ is the number of labeled nodes per class. For example, Cora-1 denotes the classification task on dataset Cora with one seed per class.

### 5.3 IMPLEMENTATION DETAILS

For all the models(Perozzi et al., 2014; Tang et al., 2015; Grover & Leskovec, 2016; Wang et al., 2016; Bojchevski & Günnemann, 2018; Velickovic et al., 2018; Monti et al., 2017) except for GCN based methods, settings of hyper-parameters are the same as suggested in original papers. All GCN based methods including GCN, Self-training GCN, Co-training GCN, Intersection GCN, Union GCN, and DSGCN share the same setting of hyper-parameter following (Shchur et al., 2018): one hidden layer with 64 units, dropout rate 0.8, Adam optimizer (Kingma & Ba, 2015) with learning rate $10^{-2}$, a $L_2$ regularization with weight $10^{-3}$. We train other GCN based methods for a fixed epochs of 200, while DSGCN is trained for 600 epochs in few-label tasks such as 1, 3, 5, 10 tasks. Because 20 or 50 labels per class implies ample supervised information, we train DSGCN for 200 epochs in these tasks. The four variants of (Li et al., 2018): Self-training GCN, Co-training GCN, Intersection GCN and Union GCN follow original self-training settings in (Li et al., 2018). For DSGCN, we use a threshold of 0.6 when the number of labels per class is below 3, and set the threshold to 0.75 for label rate above 3 but below 10. Otherwise, the threshold is 0.9 by default.

### 5.4 RESULT ANALYSIS

The numerical results are summarized in Table 2 and Table 3. The highest accuracy in each column is highlighted in bold and the top 3 are underlined. We group all models into three categories: GNN variants(GCN, GAT, MoNet), unsupervised embedding methods (DeepWalk, DGI, LINE, G2G) and GCN with self-training (Co-training, Self-training, Union and Intersection, DSGCN).

Table 2: Summary of results in terms of mean classification accuracy (in percent) over 100 random splits in different tasks. Unsupervised approaches first learn a lower-dimensional embedding for each node in an unsupervised manner, and then the embeddings are used to train a supervised classifier for node classification. Here we use logistic regression as the classifier for unsupervised embeddings.

| # of Labels | Citeseer | | | | | | Cora | | | | | |
|---|---|---|---|---|---|---|---|---|---|---|---|---|
| | 1 | 3 | 5 | 10 | 20 | 50 | 1 | 3 | 5 | 10 | 20 | 50 |
| **LP** | 30.1 | 37.0 | 39.3 | 41.9 | 44.8 | 49.5 | 51.5 | 60.5 | 62.5 | 64.2 | 67.3 | 71.7 |
| **DeepWalk** | 28.3 | 34.7 | 38.1 | 42.0 | 45.6 | 50.7 | 40.4 | 53.8 | 59.4 | 65.4 | 69.9 | 74.2 |
| **LINE** | 28.0 | 34.7 | 38.0 | 43.1 | 48.5 | 54.6 | 49.4 | 62.6 | 63.4 | 71.1 | 74.0 | 76.5 |
| **G2G** | 45.1 | 56.4 | 60.3 | 63.1 | 65.7 | 68.2 | 54.5 | 68.1 | 70.9 | 73.8 | 75.8 | 77.0 |
| **DGI** | 46.1 | 59.2 | 64.1 | **67.6** | 68.7 | 72.3 | 55.3 | 70.9 | 72.6 | 76.4 | 77.9 | 78.7 |
| **GCN** | 36.4 | 50.3 | 57.5 | 63.2 | 68.8 | 72.2 | 42.4 | 61.6 | 68.4 | 75.1 | 80.2 | 83.5 |
| **GAT** | 32.8 | 48.6 | 54.9 | 60.8 | 68.2 | 71.5 | 41.8 | 61.7 | 71.1 | 76.0 | 79.6 | 83.4 |
| **MoNet** | 38.8 | 52.9 | 59.7 | 64.6 | 66.9 | 69.9 | 43.4 | 61.2 | 70.9 | 76.1 | 79.3 | **83.9** |
| **Co-training** | 36.7 | 49.0 | 55.0 | 60.7 | 65.9 | 70.0 | 53.1 | 65.7 | 70.2 | 73.8 | 78.7 | 82.5 |
| **Self-training** | 34.6 | 50.0 | 58.7 | 67.4 | 69.1 | 71.3 | 40.6 | 63.9 | 71.1 | 75.5 | 79.1 | 81.6 |
| **Union** | 37.2 | 50.8 | 55.9 | 64.4 | 67.5 | 70.6 | 50.1 | 67.3 | 72.5 | 76.2 | 79.8 | 82.4 |
| **Intersection** | 35.3 | 51.8 | 60.7 | 67.1 | 70.2 | 72.2 | 43.1 | 64.4 | 69.5 | 73.1 | 78.4 | 82.0 |
| **DSGCN** | **53.2** | **63.9** | **65.8** | **67.6** | **70.5** | **72.4** | **62.5** | **72.3** | **75.5** | **77.7** | **80.8** | 83.8 |

Table 3: Summary of results in terms of mean classification accuracy(in percent) over 100 random splits in different tasks. GNN variants are excluded due to limited computation resources.

| # of Labels | Pubmed | | | | | | Cora-full | | | | | |
|---|---|---|---|---|---|---|---|---|---|---|---|---|
| | 1 | 3 | 5 | 10 | 20 | 50 | 1 | 3 | 5 | 10 | 20 | 50 |
| **LP** | 55.7 | 61.9 | 63.5 | 65.2 | 66.4 | 67.5 | 26.3 | 32.4 | 35.1 | 38.0 | 41.0 | 46.0 |
| **GCN** | 41.3 | 54.9 | 63.6 | 71.2 | 77.8 | **81.0** | 26.4 | 42.8 | 49.3 | 54.4 | 61.2 | **65.4** |
| **Co-training** | 55.1 | 64.7 | 69.0 | 73.5 | 77.9 | 80.5 | 28.3 | 38.1 | 42.8 | 48.5 | 53.8 | 62.2 |
| **Self-training** | 49.7 | 62.7 | 67.2 | 70.6 | 76.5 | 79.3 | 28.7 | 43.6 | 48.9 | 53.4 | 60.8 | 64.4 |
| **Union** | 55.1 | 65.4 | 69.7 | 74.0 | **78.5** | 80.9 | 29.2 | 43.3 | 48.4 | 52.9 | 59.2 | 62.2 |
| **Intersection** | 52.7 | 63.4 | 67.8 | 70.6 | 75.9 | 79.0 | 26.8 | 37.7 | 44.4 | 51.5 | 58.4 | 62.1 |
| **DSGCN** | **55.8** | **67.1** | **70.2** | **74.7** | 77.8 | **81.0** | **30.9** | **45.6** | **51.3** | **57.5** | **61.4** | 64.8 |

**Comparison Between GNN Variants and Embedding Methods**   As unsupervised methods, G2G and DGI outperform all GNN variants in very few labels cases, e.g., 1 and 3 per class on both Cora and Citeseer. Observing that LP performs well in Cora-1 while other feature propagation methods not, we can naturally conclude that in dataset with graph structure, concentrating more on the unsupervised information (both strong manifold structure(Li et al., 2018) and feature patterns) will improve semi-supervised model compared to just utilizing supervised information, in the case of low label rate. When label rate goes higher, all GNN variants enjoy better accuracies compared to unsupervised models. Hence we empirically verify the strong generalization ability of GNNs when the supervised information is sufficient. Sun et al. (Sun et al., 2019) has demonstrated the limitation of GCN in few labels case, and here we find that these convolution based methods suffer from inefficient propagation of label information as well, which can be seen as the intrinsic drawbacks of semi-supervised graph convolution based methods.

**Comparison Between Self-training GCNs and All Other Models**   In all few-label tasks, self-training strategies improve over GCN by a remarkable margin. Except for tasks with 50 labels per class, the best accuracy is always obtained by self-training GCN. Even in extreme one-label case, where unsupervised information is more vital, DSGCN outperforms G2G by a margin of 6.2% in Cora and 9.2% in Citeseer. We conclude that self-training strategy is capable of utilizing unsupervised information more effectively. Thus it significantly helps classification. Additionally, four naive self-training GCNs implemented in (Li et al., 2018) are worse than GCN when label rate goes higher, e.g., Cora-50 and Cora-full-5, which manifests that inappropriate self-training strategies will sometimes degrade the performance of the base model. Hence there is a trade-off: capturing unsupervised signals, or learning supervised information well. However, DSGCN holds a good balance here. It doesn't show much decrease compared to GCN even in the worst case task, Cora-full-50, where the

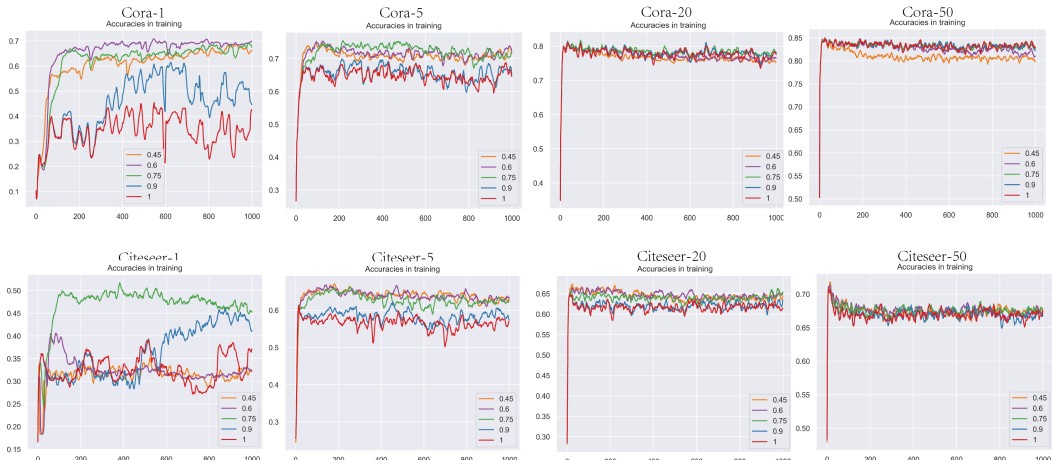

Figure 1: Test accuracies in training process. Models with different threshold are denoted with different colors, which can be distinguished in legend. Specifically, threshold 1 represents that the model is equal to original GCN.

accuracy only decreases by $0.6\%$; in all other cases it is always better than GCN. This demonstrates that the dynamic self-training framework not only helps the original model to capture unsupervised information, but also retains the learning ability when there are enough labels.

**Comparison of Self-training GCNs**   By applying a simpler and more general self-training strategy, DSGCN outperforms other self-training based GCNs with considerable margins in most cases. In Citeseer-1, the margin even reaches $14.1\%$ compared with the best strategy among Co-training, Self-training, Union and Intersection. This empirically supports the advantage of DSGCN for tackling a wide range of classification tasks over conventional self-training methods.

**Effect of Threshold**   Here we discuss how the important hyper-parameter $\beta$ influence the performance of DSGCN. We train DSGCN with different threshold: 0.45, 0.6, 0.75, 0.9, 1.0 for 1000 epochs on dataset Cora and Citeseer for the same split with the same initialized weights. We conduct these experiments on tasks with different seed numbers, the results are presented in figure 1. As shown in figure 1, when labels are very few, DSGCN with a relatively lower threshold $\beta$ demonstrate a clear improvement in accuracy over the original GCN. Besides, GCN's accuracy curve erratically fluctuates while the curve of DSGCN with a low threshold does not. Thus, we observe that the stability of the base model is also improved by wrapping it into the dynamic self-training framework. When more labels are provided, all models tend to be stable and a low threshold could harm the training process.

## 6  CONCLUSION

In this paper, we firstly introduce a novel self-training framework. This framework generalizes and simplifies prior work, providing customizable modules as extension for multi-stage self-training. Then we instantiate this framework based on GCN and empirically compare this model with a number of methods on different dataset splits. Result of experiments suggests that when labels are few, the proposed DSGCN not only outperform all previous models with noticeable margins in accuracy but also enjoy better stability in the training process. Overall, the Dynamic Self-training Framework is powerful for few-label tasks on graph data, and provides a novel perspective on self-training techniques.

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

## A  APPENDIX: ADDITIONAL EXPERIMENTS

We also test our self-training methods on other GNNs as well, e.g., SGC(Wu et al., 2019), GAT (Velickovic et al., 2018), and GraphSage (Hamilton et al., 2017). For the three GNN models, settings of hyper-parameters are the same as suggested in original papers. And our dynamic self-training framework share the same setting of hyper-parameter: one hidden layer with 32 units, dropout rate 0.7, Adam optimizer (Kingma & Ba, 2015), a $L_2$ regularization with weight $5^{-4}$ and set the threshold to 0.9. Clearly, our dynamic self-training framework achieves similar improvements on all the three base models. The numerical results are summarized in Table 4. We can see equipped with our DS framework, these models enjoys noticeable increase in performance.

Table 4: Summary of results in terms of mean classification accuracy (in percent) over 50 random splits in different tasks(the results of GAT experiments are from Table 2).

|  | Citeseer | | | | Cora | | | |
|---|---|---|---|---|---|---|---|---|
| # of Labels | 5 | 10 | 20 | 50 | 5 | 10 | 20 | 50 |
| **SGC** | 55.5 | 63.7 | 69.0 | 72.6 | 63.5 | 72.5 | 75.9 | 78.9 |
| **DS-SGC** | 59.6 | 65.0 | 69.7 | 73.4 | 65.0 | 73.4 | 76.2 | 78.9 |
| **GAT** | 54.9 | 60.8 | 68.2 | 71.5 | 71.1 | 76.0 | 79.6 | 83.4 |
| **DS-GAT** | 58.3 | **67.0** | **70.8** | **73.4** | 71.9 | 77.1 | **81.0** | 83.6 |
| **GraphSAGE** | 59.7 | 65.4 | 68.8 | 72.1 | 69.3 | 75.3 | 79.2 | 82.5 |
| **DS-GraphSAGE** | **60.6** | 66.3 | 69.5 | 72.6 | **72.5** | **78.4** | **81.0** | **84.0** |

To evaluate the computation overhead introduced by dynamic self-training framework, we test the total training time for various models. Intuitively the computational cost will only slightly increase. The reason is that the computational cost of the original GCN model is dominated by previous layers, where the entire graph is included. So even if all nodes become pseudo labels, the size of the entire network is increased by at most a factor of 2, and the number of parameters remains the same. Therefore, the computational costs will increase by at most a small constant in theory. We have also verified this empirically. We record the training time of base models before and after applying our framework. In the experiments, the training size is 20 per class, the number of epoch is 200, and the time is the average time (in seconds) of 25 runs. The numerical results can be seen in Tabel 5.

Table 5: Total training time for various models in seconds(s), implemented on PyG.

|  | Citeseer | Cora |
|---|---|---|
| **GCN** | 3.0 | 2.6 |
| **DSGCN** | 9.7 | 6.3 |
| **SGC** | 1.4 | 1.3 |
| **DS-SGC** | 7.1 | 7.4 |
| **GAT** | 5.2 | 4.7 |
| **DS-GAT** | 11.9 | 8.7 |
| **GraphSAGE** | 1.9 | 2.1 |
| **DS-GraphSAGE** | 8.7 | 8.5 |

