# OpenReview forum: "DYNAMIC SELF-TRAINING FRAMEWORK  FOR GRAPH CONVOLUTIONAL NETWORKS"
_ICLR.cc/2020/Conference — Reject_

### Official Review · AnonReviewer3 · 2019-10-23
**Official Blind Review #3**

**Rating:** 3

**Review:**

This paper propose to modify the existing work [1] of self-training framework for graph convolutional networks. It tracks three limitations of [1] and propose three  use a threshold-based rule to insert new pseudo-labels and dynamic change the pseudo-label set. Moreover personalized weight are assigned to each activate pseudo-label proportional to its current classification margin. Evaluation of the proposed framework is performed on four networks for semi-supervised node classification task with varying label rates.
Pros:
1. This work tracks and addresses the limitations of existing work.
2. Authors conduct experiments on multiple dataset with varying 2-hop coverage ratio.
3. The overall paper is well written, except some typos, e.g. in page 6, section 5.1 "Each of three dataset is ......". Should "three" be "four".
Cons:
1. The proposed framework makes modification on the existing work, which is a good extension but the novelty is limited.
2. The gap of the experiment results between the proposed method and the baseline methods are quite small.
3. Only GCN instantiation are provided, it is suggested to evaluate the effectiveness on the other GNN variants, such as GraphSage, GAT and MoNet.
[1] Li et al. Deeper insights into graph convolutional networks for semi-supervised learning.

**Experience Assessment:**

I have published in this field for several years.

**Review Assessment: Checking Correctness Of Derivations And Theory:**

I carefully checked the derivations and theory.

**Review Assessment: Checking Correctness Of Experiments:**

I carefully checked the experiments.

**Review Assessment: Thoroughness In Paper Reading:**

I read the paper thoroughly.

---

> ### Author Response · Authors · 2019-11-15
> **Response to AnonReviewer #3**
>
> We thank the reviewer for the constructive comments.
>
> 1. "Only GCN instantiation are provided, it is suggested to evaluate the effectiveness on the other GNN variants, such as GraphSage, GAT and MoNet."
>
> Thanks for the suggestion! We have implemented and tested our self-training methods on other GNNs as well, e.g., GraphSage, GAT, SGC, and some preliminary results are as follows.
> ---------------------------------------------------------------------------------------------------
>                                                   Cora
> ---------------------------------------------------------------------------------------------------
>                               5            10              20             50
> ---------------------------------------------------------------------------------------------------
> GraphSage       69.3         75.3          79.2          82.5
> DSGraphSage  72.5         78.4          81.0          84.0
> ---------------------------------------------------------------------------------------------------
> GAT                    71.1         76.0          79.6          83.4
> DSGAT               71.9         77.1          81.0          83.6
> ---------------------------------------------------------------------------------------------------
> SGC                    63.5         72.5         75.9          78.9
> DSSGC               65.0         73.4         76.2          78.9
> ---------------------------------------------------------------------------------------------------
>
> ---------------------------------------------------------------------------------------------------
>                                              Citeseer
> ---------------------------------------------------------------------------------------------------
>                                5             10           20             50
> ---------------------------------------------------------------------------------------------------
> GraphSage        59.7        65.4         68.8          72.1
> DSGraphSage   60.6        66.3         69.5          72.6
> ---------------------------------------------------------------------------------------------------
> GAT                     54.9        60.8         68.2          71.5
> DSGAT                58.3        67.0         70.8          73.5
> ---------------------------------------------------------------------------------------------------
> SGC                     55.5        63.7         69.0           72.6
> DSSGC                59.6        65.0         69.7           73.4
> ---------------------------------------------------------------------------------------------------
> Clearly, our dynamic self-training framework achieves similar improvements on all the three base models. For these models, we observe notable improvements even when the label rate is high. We emphasize that our framework is very flexible to use: to apply to a different a base model, the implementation and tuning is almost effortless. All the additional experimental results will be added in the final version.
>
> 2. "The proposed framework makes modification on the existing work, which is a good extension, but the novelty is limited."
>
> We don’t think our framework is a just modification of [1].
>
> We propose three techniques to address additional complexity introduced when applying a standard self-training framework. Our techniques are very effective in this aspect: only roughly 15 lines of code are needed to wrap any base GNN model, only one additional hyperparameter, and significant accuracy boosting even compare with standard self-training.
>
> Moreover, in our framework, we update the training set after each epoch instead of after a full training of the base model, and we also propose a simple gradient computation strategy, so that the training process is end-to-end and there are no explicit training stages as in standard multistage self-training. This makes the model easier to implement, train, and extend.
>
> Our framework's apparent simplicity might make it seem like a simple modification, but we don’t think simple means limited novelty. Simplicity, flexibility, and better performance of our framework makes it more suitable to use in practice.
>
> 3. "The gap of the experiment results between the proposed method and the baseline methods are quite small."
> This is not true. 1. The improvement achieved by dynamic self-train over the base model GCN could be more than 15%. 2. Compared with the four self-training methods in [1], the gap could be also as large as 14%.
>
> [1] Li et al. Deeper insights into graph convolutional networks for semi-supervised learning.

---

### Official Review · AnonReviewer1 · 2019-10-23
**Official Blind Review #1**

**Rating:** 6

**Review:**

The paper proposes an approach for learning graph convolutional networks for inferring labels on the nodes of a partially labeled graph  when only limited amount of labeled nodes are available.

The proposal is inspired from Graph convolution Networks with the idea of overcoming the major drawback of these models that lies of their behavior in case of limited coverage of the labeled nodes, which implies using deeper versions of the model leading at the price of what the authors call the over-smoothing problem.

The main idea here consists in relying on self training to get a better coverage of labeled nodes enabling learning with less deep models, this translates to a simple and intuitive algorithm. Using self training is not new in GCN but the way it is used here, computing adaptively a threshold for incorporating pseudo labels and using weights according to the confidence off predictions is new.

Experimental results are reported on citation datasets and compared with many baselines show similar results as baselines when the coverage increases up to 50 labeled nodes /class, but the method brings significant improvements when the coverage is low (e.g. only few, <20, labels /class).

Although the difference with previous approaches do not look like a huge step, the method seems to be quite justified empirically and achieve real good results wrt state of the art.

**Experience Assessment:**

I have read many papers in this area.

**Review Assessment: Checking Correctness Of Derivations And Theory:**

I carefully checked the derivations and theory.

**Review Assessment: Checking Correctness Of Experiments:**

I assessed the sensibility of the experiments.

**Review Assessment: Thoroughness In Paper Reading:**

I read the paper thoroughly.

---

> ### Author Response · Authors · 2019-11-15
> **Response to AnonReviewer #1**
>
> We thank the reviewer for the constructive comments.
>
> We have revised our paper. As suggested by the reviewers, additional experiments are conducted. We have tested the computation overhead of applying our dynamic self-training framework. The results show that the training time increases only slightly. Moreover, we test the effect of our framework being applied on other GNN models, including GraphSage, GAT, and SGC. We observe notable improvements on all these models. Please see our responses to other reviewers for more details on the experimental results.

---

### Official Review · AnonReviewer2 · 2019-10-28
**Official Blind Review #2**

**Rating:** 6

**Review:**

#Summary

This paper proposes a generalised self-training framework to build a Graph Neural Network to label graphs.  Of importance is the dynamic nature of the self-training. The authors do not change the GCN but extend the self-training portion as per the prior GCN paper by introducing Dynamic Self-Training that keeps a confidence score of labels predicted for unlabelled nodes.

# Comments

This is a very interesting paper in terms of looking at the effects of changing the self-training framework to better utilise the underlying structure. As such we can exploit information from other nodes that are yet to be labelled.

1. As the self-training is going on, are there different computational costs or are they about the same?
2. For CiteSeer 20 and 50, why does \beta = 0.45 switch from the other experiments?
3. Will such self-training be useful for general NN self-training procedures
4. If we had soft-labelling or uncertainty on which label each node has, how would the dynamic self-training be changed?

#Other notes
Please remove the

An appendix
You may include other additional sections here

**Experience Assessment:**

I do not know much about this area.

**Review Assessment: Checking Correctness Of Derivations And Theory:**

I assessed the sensibility of the derivations and theory.

**Review Assessment: Checking Correctness Of Experiments:**

I carefully checked the experiments.

**Review Assessment: Thoroughness In Paper Reading:**

I read the paper at least twice and used my best judgement in assessing the paper.

---

> ### Author Response · Authors · 2019-11-15
> **Response to AnonReviewer #2**
>
> We thank the reviewer for the constructive comments.
>
> 1. "As the self-training is going on, are there different computational costs or are they about the same?"
> The computational costs is only slightly increased. The reason is that the computational cost of the original GCN model is dominated by previous layers, where the entire graph is included. So even if all nodes become pseudo labels, the size of the entire network is increased by at most a factor of 2, and the number of parameters remains the same. Therefore, the computational costs will increase by at most a small constant in theory. We have also verified this empirically. We record the training time of GCN and GAT before and after applying our framework. In the experiments, the training size is 20 per class, the number of epochs is 200, and the time is the average time (in seconds) of 25 runs.
> ---------------------------------------------------------------------------------------------------
> Cora
>            GCN        2.6
>          DSGCN     6.3
>
>            GAT        4.7
>          DSGAT     8.7
> ---------------------------------------------------------------------------------------------------
>
> ---------------------------------------------------------------------------------------------------
>     Citeseer
>           GCN         3.0
>          DSGCN     9.7
>
>          GAT           5.2
>         DSGAT      11.9
> ---------------------------------------------------------------------------------------------------
>
> 2. "For CiteSeer 20 and 50, why does \beta = 0.45 switch from the other experiments?"
> We didn’t use \beta = 0.45 for CiteSeer 20 and 50. As explicitly explained in the paper, we use a threshold of 0.6 when the number of labels per class is below 3 and set the threshold to 0.75 for label rate above 3 but below 10. Otherwise, the threshold is 0.9 by default. In Figure 1, \beta=0.45 here is used for comparison with other thresholds on various label rate. We are sorry for any ambiguity in the paper.
>
>
> 3. "Will such self-training be useful for general NN self-training procedures"
> We believe our self-training method will be useful for general NN training, although we think it is most effective for GNN models. We have tested our self-training methods on other GNNs, e.g., GraphSage, GAT, SGC, and some preliminary results are as follows.
> ---------------------------------------------------------------------------------------------------
>                                                   Cora
> ---------------------------------------------------------------------------------------------------
>                               5            10              20             50
> ---------------------------------------------------------------------------------------------------
> GraphSage       69.3         75.3          79.2          82.5
> DSGraphSage  72.5         78.4          81.0          84.0
> ---------------------------------------------------------------------------------------------------
> GAT                    71.1         76.0          79.6          83.4
> DSGAT               71.9         77.1          81.0          83.6
> ---------------------------------------------------------------------------------------------------
> SGC                    63.5         72.5         75.9          78.9
> DSSGC               65.0         73.4         76.2          78.9
> ---------------------------------------------------------------------------------------------------
>
> ---------------------------------------------------------------------------------------------------
>                                              Citeseer
> ---------------------------------------------------------------------------------------------------
>                                5             10           20             50
> ---------------------------------------------------------------------------------------------------
> GraphSage        59.7        65.4         68.8          72.1
> DSGraphSage   60.6        66.3         69.5          72.6
> ---------------------------------------------------------------------------------------------------
> GAT                     54.9        60.8         68.2          71.5
> DSGAT                58.3        67.0         70.8          73.5
> ---------------------------------------------------------------------------------------------------
> SGC                     55.5        63.7         69.0           72.6
> DSSGC                59.6        65.0         69.7           73.4
> ---------------------------------------------------------------------------------------------------
>
>
> 4. "If we had soft-labelling or uncertainty on which label each node has, how would the dynamic self-training be changed?"
> In general, we can treat all original labels as “pseudo labels” as well. We just need a mechanism to determine the initial confidence of these labels and then all labels in the training set can be treated equally.

---

### Decision · Program_Chairs · 2019-12-19

**Decision:**

Reject

**Comment:**

The paper is develops a self-training framework for graph convolutional networks where we have partially labeled graphs with a limited amount of labeled nodes. The reviewers found the paper interesting. One reviewer notes the ability to better exploit available information and raised questions of computational costs. Another reviewer felt the difference from previous work was limited, but that the good results speak for themselves. The final reviewer raised concerns on novelty and limited improvement in results. The authors provided detailed responses to these queries, providing additional results.

The paper has improved over the course of the review, but due to a large number of stronger papers, was not accepted at this time.